# Cost-effectiveness analysis of adding tuberculosis household contact investigation on passive case-finding strategy in Southwestern Uganda

**Dickens Odongo** [1]*, **Bernard Omech** [1], **Alfred Acanga** [2]

**1** Department of Environmental Health and Disease Control, Faculty of Public Health, Lira University, Lira, Uganda, **2** Faculty of Management Sciences, Lira University, Lira, Uganda

* dodongo1980@gmail.com

## Abstract

### Introduction

The standard passive case-finding strategy implemented by most developing countries is inadequate to detect new cases of Tuberculosis. A household contact investigation is an alternative approach. However, there is limited cost-effectiveness data to support planning and implementation in low and middle-income countries. The study aimed to evaluate the cost-effectiveness of adding household contact investigation (HCI) to the passive case-finding (PCF) strategy in the Tuberculosis control program in Southwestern Uganda.

### Methods

We conducted an economic evaluation using a retrospective study approach and bottom-up costing (ingredients) techniques. It was a synthesis-based evaluation of existing data extracted from the District Health Information System (DHIS 2), TB registers, and a primary cost survey. The study compared two methods of Tuberculosis (TB) case finding (PCF and HCI) strategies. Regarding PCF, patients either self-reported their signs and symptoms or were prompted by healthcare workers. At the same time, HCI was done by home visiting and screening contacts of TB patients. Patients and household contacts presumed to have Tuberculosis were requested to produce samples for analysis. We applied a static decision-analytic modeling framework to examine both strategies' costs and effectiveness. The study relied on cost and probability estimates from National Tuberculosis (TB) program data, activity costs, and published literature. It was performed from the societal and provider perspectives over 1.5 years across 12 facilities in Ntungamo, Sheema, and Rwampara Districts. The primary effectiveness measure was the number of TB cases detected (yield) and the number needed to screen (NNS). The TB yield was calculated from the number of patients screened during the period under study. The incremental cost-effectiveness ratio (ICER) was expressed as cost in 2021 US$ per additional TB case detected. We did not apply a discount rate because of the short analytic time horizon.

**Data Availability Statement:** All relevant data are within the paper and its Supporting information

**Funding:** The author(s) received no specific funding for this work.

**Competing interests:** The authors have declared that no competing interests exist.

## Results

The unit costs of detecting a Tuberculosis case were US$ (United States dollar) 204.22 for PCF and US$ 315.07 for HCI. Patient and caregiver costs are five times more in PCF than in HCI [US$26.37 Vs. US$ 5.42]. The ICER was US$ 3,596.94 per additional TB case detected. The TB screening yields were **0.52%** (1496/289140) for passive case finding and **5.8%** (197/3414) for household contact investigation. Household contact investigation yield among children 0–14 Vs. 15+ years [6.2% Vs.5.4%] *P = 0.04*. The Yield among People living with HIV (PLHIV) Vs. HIV-negative [15.8% Vs.5.3%] *P* = 0.03 in HHCI. The PCF yield in men Vs. Women [1.12% Vs.0.28%] *P*<0.01. The NNS in PCF was **193** [95% CI: 186–294] and **17** [95% CI: 14–22] in HCI.

## Conclusion

Our baseline assumptions and the specific implementations of adding HCI to existing PCF programs in the context of rural African settings prove to be not cost-effective, rather than HCI as a strategy. HCI effectively identifies children and PLHIV with TB and should be prioritized. Meanwhile, the Passive case-finding strategy effectively finds men with TB and costs lower than household contact investigation.

## Background

Tuberculosis (TB) remains a significant clinical and public health problem, despite being largely curable and preventable. Globally, 2.9 million of the estimated 10.0 million new TB cases were missed in 2019 [1]. Of the reported 7.1 million people newly diagnosed with TB in 2019, South-Eastern Asia accounted for more than half (3,641,245), followed by Africa and Western Pacific regions at 1,436,330 and 1,416,592, respectively. The gap could be due to under-reporting, underdiagnosed, and inadequate access to healthcare by most people in high TB burden countries. Uganda belongs to one of the World Health Organization's (WHO) 30 high TB and TB/HIV burden countries. In 2019, the country notified only 65,900 of the estimated 88,000 new TB cases; of the 65,900 notified, 15,600 people who fell ill with TB died [1].

As a result of low case detection and high death rates in countries with high TB and TB/HIV burdens, WHO launched the End TB Strategy in 2015 to end the global TB epidemic by 2035. These ambitious targets aim to reduce TB incidence by 90% and death by 95% [2]. Traditionally, the National TB programs have relied on a standard passive TB case-finding strategy where patients present themselves to the healthcare facility for TB evaluation and diagnosis [3]. PCF has had limited success in Africa, casting doubts over attaining the End TB strategy targets. The primary reasons include patients 'delays, a lack of awareness of TB symptoms, and healthcare access [4]. In addition, in those seen by PCF, the patients experience long delays before diagnosis, thus transmitting the disease while still in the community [5]. Moreover, high out-of-pocket costs and treatment for PCF have proved catastrophic for many people [6]. Hence, it is imperative to add strategies that improve TB case finding in settings and populations with poor access to and uptake of TB diagnosis and care.

In 2012, the WHO recommended contact investigation for household contacts of patients with bacteriologically confirmed TB in low and middle-income countries [7]. It entails screening for Tuberculosis disease, with or without latent TB infection, among contacts at the household level. Although the WHO and Uganda National guidelines recommend household contact tracing, implementation has been challenging due to limited available evidence for

effectiveness or cost-effectiveness [8]. In addition, limited data are available for the additional cost of implementing household contact investigation, primarily when implemented under routine program conditions. A study from Malaysia reported the cost of active contact investigation to be US$6.60 per single contact tracing to visit with a yield of 0.5% [9]. In Peru, adding active contact tracing to PCF incurred an incremental cost of US$48.8 to evaluate household contacts of an index TB patient. The incremental cost-effectiveness ratio (ICER) of US$1811 per Disability Adjusted Life Year (DALY) was averted [10]. Few studies have compared the effectiveness and costs of adding Household contact investigation to passive case findings under programmatic conditions in Uganda [11, 12]. Policymakers and program staff face significant challenges in measuring the effectiveness of newly introduced interventions and reviewing the feasibility of scaling up successful approaches [13]. The cost of household contact investigation, including additional efforts required by the already stretched healthcare system, has often been cited as challenging its execution [14]. Moreover, the requirement to reallocate resources to identify, diagnose and treat people with TB symptoms among competing priorities in primary care settings has been challenging [15].

Despite multiple regional partner interventions, the previous data show low TB case detection in Southwestern Uganda [16]. The region notified only 94.4/100,000 compared to the national performance of 118.7/100,000. It detected only 66.25% of TB cases in 2019 against 90% of the End TB strategy due to missed or delayed diagnoses. The program attributes low case detection to low yield and quality of screening in household contact investigations and passive case finding. In addition, other factors, including poor access to TB diagnostics, inadequate skills and knowledge about TB among healthcare workers, and inadequate resources allocated to TB programs, played a critical role [17]. In five of the eighteen Districts of Southwestern Uganda, the Elizabeth Pediatric AIDS Foundation (EGPAF) two projects, the Catalyzing Pediatric Tuberculosis Innovation (CaP-TB) funded by the Innovation in Global Health (Unitaid) and the Regional Health Integration to Enhance Services in Southwest (RHITES-SW) funded by the United States Agency for International Development (USAID) partnered with the Ministry of Health's National Tuberculosis and Leprosy Program (NTLP) to improve TB screening, diagnosis, and treatment between 2017 to 2021. The projects supported integrating and decentralizing pediatric and adult TB services and screening at health facilities and the community levels. The CaP-TB project recruited and built the capacity of TB linkage facilitators and healthcare workers to improve the quality and effectiveness of TB services along the cascade of care at the facility and community levels. The districts were supported to plan and systematically screen for TB at all entry care points, among household contacts and known TB hotspots in the community. The study aimed to evaluate the cost and the incremental cost-effectiveness ratio (ICER), the TB yield, and the number needed to screen from passive case finding (PCF) and household contact investigation (HCI) in Southwestern Uganda during the period under study.

## Methods

### Study design, area, and settings

The study utilized decision-analytic modeling as a quantitative framework to synthesize TB screening and diagnostic cascades. Data were extracted retrospectively from the District Health Information System (DHIS 2), contact tracing, and unit TB registers. The evaluation considered healthcare providers and societal study cost perspectives. Data on costs included healthcare providers 'experiences through interviews and patients through a cost survey.

We evaluated 12 facilities in the three districts of Ntungamo, Rwampara, and Sheema in the Ankole sub-region, Southwestern Uganda, with an overall mid-year population of 920,500

[18]. The study chose the three Districts and 12 facilities purposely because of the high prevalence of TB and improved data recording and reporting quality in primary and secondary tools. In addition, the three Districts were consistently supported in planning and implementing household contact investigations compared to other Districts from Southwestern Uganda. The combined population translates to an annual estimated TB notification incidence of 1,767 (192/100,000) new and relapse persons [17]. There are 33 Diagnostic TB and treatment units, including two general hospitals, eight Health Center (HC) IVs, 22 HC IIIs, and 1 HC IIs. By Ministry of Health standard guidelines, a district hospital serves 500,000 people. In comparison, HC IV and HC III serve 100,000 and 20,000 people, respectively.

## Study population and period

The study targeted children and adults in the three Districts served by 920,500 people, 12 Diagnostic TB and treatment units (DTUs), and consent healthcare providers involved in TB diagnosis, treatment, and prevention. In addition, we included all consented and presumptively diagnosed patients from Passive and Household contact investigation activities between January 2020 and June 2021. The study excluded data on transfer in TB patients or diagnosed from other districts since they would not form part of the denominator for cost estimation and geographical case finding; TB index cases without contact tracing and passive contact tracing data, and those who were ill or bedridden did not participate in the cost survey study. Furthermore, we excluded children (< 18 years) from the cost survey because they could not provide accurate data and information on cost estimates. However, we included children 0–14 years and those above 15 in other analyses and results presentations. In Uganda, the household contact investigation strategy would mainly find children under 15 years old because they will likely be at home with their parents [19].

## Study procedure, case definition, and data collection

The researcher evaluated two alternative case-finding strategies, Passive case finding (PCF) alone and Passive case finding (PCF) plus Active Household Contact Investigation (HCI)

**Passive case finding (PCF).** Passive case-finding (PCF) adopted by Uganda's National TB and Leprosy program is a universally acceptable WHO standard policy recommendation for TB case detection [12]. A patient with symptoms of TB initiates a visit to a health facility for clinical and diagnostic evaluation. A patient presumed to have TB is subjected to intensified Case Finding tool (ICF) for screening (the primary assessment tool for individuals with TB symptoms who self-report to health facilities for TB care). It assesses a cough lasting two or more weeks or any cough for people living with HIV (PLHIV) or persistent fever for two weeks or more, noticeable weight loss of more than 3kgs, and excessive night sweats for over three weeks. The primary assessment among children under five years was a history of poor weight gain in the last month and contact with a person diagnosed with pulmonary TB or chronic cough. Those presumed for TB cases are then referred for further evaluation to ensure a definitive diagnosis of TB. The diagnostic tools used included GeneXpert RIF/Ultra, microscopy, urine LAM or clinical diagnosis as per MoH/NTLP diagnostic algorithms for adults and children. The evaluation assumed all diagnosed cases accessed GeneXpert tests. Data recording and reporting for PCF were done through HMIS and DHIS 2 of the Ministry of Health.

**Passive case finding and Active household contact investigation (HCI).** The study evaluated a hypothetical combination of HCI under the current PCF approaches. Household contact investigation is a targeted form of active case finding that aims to identify additional Tuberculosis cases among household contacts.

The study followed five steps while conducting a Household contact investigation [20]; 1) Interview the index TB case from PCF, informing the patient about TB disease; 2) Create a contact tracing list; 3) Scheduled health workers who will conduct contact tracing, 4) Plan for the contact tracing activities, and 5) Screen contacts for TB (Home visits for household contact investigations).

Before the activity, the team prepared the following tools, materials, and logistics;(gloves, HIV Testing stripes, sputum mugs, TB contact screening forms, syringes, patient referral forms, job aids, Intensified Case Finding aide, TB/HIV health education flip chart), cotton swabs, zip lock bags, face masks, N95 masks, and ice-cool boxes. Upon reaching the home of an index case, the team offered health education to household members and sought consent. The initiation of TB screening among family members follows using ICF. Any household contact reporting signs and symptoms in the contact tracing tool was a presumptive TB case.

The health workers instructed the presumptive persons to produce 3-5mls of sputum, pack it, and send it for analysis. Health workers labeled the sputum mug with the presumptive patient's name and contacts; the results were recorded in the unit TB register.

Samples analyses for diagnosis were done through either GeneXpert MTB/RIF Ultra (Cepheid Inc. Sunnyvale, USA), microscopy, or TB LAM. Other presumptive cases that could not provide a sample during the household visits were given a referral to the facility for sample collection.

Recording and reporting active household contact investigation was done through the presumptive contact tracing and unit TB registers for confirmed cases. However, the final report sent to the Ministry of Health (MoH) through DHIS2 had challenges reporting active household contact tracing data because of the knowledge gap among healthcare workers and synchronization between primary and secondary data recording and reporting tools. The researcher used primary records (contact tracing and unit TB registers) to capture data from Jan 1, 2020, to Jun 30, 2021.

Data collection relied on the Health Management Information System (HMIS) tools, DHIS 2, payrolls, and interviews with patients and healthcare providers. Through the HMIS, the Ministry of Health reports, the national TB program collects data on the number of people screened, tests performed, and cases diagnosed. The researcher extracted and exported the needed data into Excel (Microsoft version 19) for PCF from Jan 1, 2020, to Jun 30, 2021.

The researcher used a contact tracing investigation tool [S6 File] to extract data from the TB contact tracing register for active household contact investigations. The following variables on index patients and contacts were collected. The TB number, age, sex, Number of contacts screened, Number of contacts presumed for TB, Number of contacts diagnosed with TB, the number offered HIV tests, and number found HIV positive.

We used a structured questionnaire [S3 File] to collect data on the TB patients and their caregivers 'costs (direct and indirect costs) for each diagnosis-related visit during the evaluation period until the initiation of TB diagnosis treatment. We define the direct costs as out-of-pocket expenditures, including transportation fees, food, and drinks for the patients and their caregivers during TB diagnosis evaluation. Indirect costs include travel time, waiting time, and absence from work by the patient or caregivers. The study estimated caregivers 'time when a family member or a friend escorted the patients to the outpatient clinic visits.

Additionally, we interviewed 12 healthcare providers (1 from each facility) using a semi-structured interview guide [S5 File] to estimate program costs for each intervention. The Program costs were incurred at the administrative levels outside the delivery and personnel costs [21]. Medical expenses included all costs for delivering health care, such as tests, drugs, and outpatient visits [22]. Finally, the researcher reviewed payment vouchers, activity requests, and payroll from the Office of the Chief Administrator Officer's Officer. The verification helped

obtain additional information on healthcare workers' salaries and training and varied information from health care providers 'interviews.

## Sample size and sampling procedure for cost survey and health care provider interview

The sample size for the cost survey was estimated using a standard formula by Leslie Kish (1965) [23]. The researcher arrived at a sample size of 45 patients for the cost survey. The figure was adjusted for missing data and the non-response rate of 50 respondents. The researcher developed a sample frame for an accessible population and employed systematic random sampling to select study participants for the cost survey.

The researcher purposively sampled 12 facilities (1 hospital, 1 HC II, 5 HC IV, and 5 HC III) from the 33 Diagnostic TB treatment units (DTU) to represent the study health facilities. The assumption was that it would give us more data points on cost estimates. The researcher clustered 12 facilities into three equal clusters. In a group, two trained research assistants randomly sampled the first subject and evenly selected study participants using a constant interval for each clinic sample frame until the total sample size of 50 for the cost survey was attained.

In addition, the researcher purposively selected 12 healthcare providers, including District TB and Leprosy supervisors (DTLS), facility TB focal persons, program officers at EGPAF Uganda, and the designated facility in charge involved in TB planning, implementation, monitoring, supervision, and evaluation team for key informant interviews. These careers and prospective approaches elicit expectations about resource use from knowledgeable TB health service delivery stakeholders.

## Data analysis, cost estimates, and data sources

We applied descriptive statistics on demographic characteristics of index and contacts to comprehend the frequencies, proportions, percentages, and appropriate measures of central tendencies (mean, median).

The yield of active TB and NNS calculation at a 95% confidence interval and the corresponding P-Value were compared for the two approaches. The yield calculation was divided by the number of TB cases diagnosed for each intervention by the number screened as the denominator. We listed all program, direct medical, and patient and caregiver costs and utilized bottom-up approach costing.

The analysis considered estimates from the program, medical, direct, and indirect costs for each intervention from the healthcare providers and societal perspectives for 18 months (Jan 1, 2020, to Jun 30, 2021) adjusted to the 2021 United States dollar exchange rate. As recommended by the panel, the consumer price index and general inflation rates in Uganda 2020 as a base year and cost-effectiveness [22] were applied. The researcher did not apply a discount rate because of the short analytic time horizon. The analysis of patient costs from HCI was excluded because healthcare providers often visited patients and their contacts from home and transmitted feedback on diagnostic outcomes. The costs of diagnosing a TB case for each intervention were subsequently summed.

The unit cost of diagnosing one TB case was estimated by dividing the cost of diagnosing a TB case by each intervention. The prices were converted from Uganda Shillings to the 2021 US Dollars (US) exchange rate (**1** US$ = **3591**UGX).

Finally, we estimated the total direct patient and caregiver costs, including out-of-pocket expenses from transportation and meals while attending TB clinic visits for diagnosis from a patient cost survey [S4 File].

**Personnel costs.**

1. For the PCF, we collected data on personnel time spent by Nurses, TB linkage facilitators, Clinicians (Clinical Officers and Medical Officers), and Laboratory Technologists. The number of samples analyzed by each carder per day and an average of twelve samples were analyzed using a time series. We calculated the hourly rates from the monthly salaries paid by the Uganda government in 2021, assuming 40 hours a working week [24]. The total personnel costs were computed by multiplying the hourly rate by the estimated patient contact time; the assumption was that every suspected TB received the GeneXpert test. Hence the Number of GeneXpert tests was used as a proxy for clients seen by the health care providers for the TB case finding.

2. For the HCI, we used a stop-clock to obtain the time it took healthcare workers to travel from the health unit to the patient's home, conduct a contact investigation, and back. The personnel cost was then estimated using the hourly pay and time for contact tracing and passive case finding. The other expenses incurred were meals and minor expenses for the healthcare workers. US$ 5.576 was given per day per person when they did perform HCI as guided by rates from the Office of the UN Resident Coordinator in Uganda [25].

**Transportation costs.** The study considered transportation costs incurred during the analysis when healthcare workers traveled to the communities to perform case-finding activities in HCI. The distance between the health facility and the index patient's home was estimated from the mileage of a motorbike or vehicle used for transport. From the findings, the healthcare workers received Uganda Shillings (UGX) 200 (US$ 0.056) per kilometer per the Office of the UN Resident Coordinator in Uganda.

**Training costs.** Training costs came from extra training of healthcare workers to perform the HCI activities. Furthermore, the study did not consider additional training costs in calculating PCF since the existing program personnel can perform routine case-finding activities at the facilities. The total training costs were obtained from project expense records and divided by the number of persons screened to obtain the per-person training cost. Two healthcare workers from each facility received five training days and were paid per diems and transport costs. The per diem was US$ 44.89 per participant and facilitator as directed by standing orders for the government of Uganda per night. Participants got transport allowance by public means and calculated depending on the distance from the residence health facility of the participant to the training venue at a rate of (UGX) 200 (US$ 0.056) per Kilometer covered. Other costs included hiring the training venue, payment for meals and refreshments during training and fuel, and per diems for facilitators.

**Communication costs.** The study considered the communication costs incurred during the HCI activities from phone communication among health workers for linkage and follow-up from facility phone records. During the study period, 4.18US$ and 6.68US$ per month were spent on PCF and HCI, respectively.

**Medical costs.** Medical costs included the GeneXpert tests and consumables (sputum cups, gloves, masks, and N95). Regardless of the case-finding strategy, we obtained the costs from the National Medical Stores (NMS) catalog, and the charges were market-based. The NMS is a government of Uganda agency responsible for supplying medicines and other related products in Uganda [26]. The literature shows that a unit cost of an MTB/RIF/Ultra GeneXpert test per patient, on average, costs US$ 21 in Uganda [27]. From the program data, the number of tests done under each approach was 11049 for PCF and 381 for HC1. The study

assumed all tests accessed GeneXpert. The quantities of consumables healthcare workers used for a single patient were calculated.

**Patient and caregiver direct costs.** The researcher estimated direct patient and caregiver costs, including out-of-pocket expenses from a patient survey [S4 File]. Patients detected through the PCF strategy incurred an average price of $18.01 because of the need to travel at least two times to the clinic before a diagnosis is confirmed. Direct caregiver costs would be like patient costs, except they were calculated based on the proportion of patients who reported having used caregivers. The HCI patients would spend $4.92; this applies only to those who need to travel to the health facility for a chest x-ray.

**Patient and caregiver indirect costs.** Patient indirect costs are estimated based on patient time spent traveling, waiting time, the diagnostic evaluation process at the health facility visit, and lost workdays. Patients incurred minimal HCI or no indirect costs since people are evaluated in their homes. From the survey results [S4 File], patients and caregivers in PCF lost 80.15 hours on average during the diagnostic evaluation process. The researcher multiplied the time lost by Uganda's minimum hourly wage of $0.15 per hour [28] to obtain the indirect cost. We valued patients' and caregivers' time using the minimum wage in Uganda as a proxy for the value of time for a person who is a non-wage earner[29].

## Effectiveness measure and data sources

The study's primary effectiveness measure was the number of actual TB cases detected and the number needed to screen. These are intermediate results of interest to healthcare workers and program implementers. In each strategy, we assigned a numeric counter (payoff) of 1 for actual TB cases detected and a zero for a true negative, false positive, or false-negative case. In addition, the PCF+HCI approach attracted a payoff of 2 for any positive case to reflect an additional Tuberculosis case detected through household contact investigation efforts. The data sources were National TB program data (HMIS) for PCF, contact tracing and unit TB registers for HCI, and published studies for HCI. The costs used in the model were those we obtained through the cost survey [S4 File], interviews, expert opinion, and literature.

## Decision model and assumptions

The study designed the decision model using TreeAge Pro2019, R1Healthcare software, to calculate the cost-effectiveness analysis. Both strategies showed the TB detection phase through presentations and diagnosis. The models for PCF alone and PCF + HCI access to the health facility. It assessed for a chronic cough, giving a sputum sample for TB diagnosis or having a chest X-ray and clinical assessment to detect TB. As demonstrated in [Fig 1] below, both case detection strategies follow the same path to the point where the TB-positive case is tracked in their household. Through a literature scan, researchers [12] assumed that just an average of potential TB suspects might access the health facility for testing. One of the study's assumptions was that 57% of the presumptive TB cases access the health facility. It is also assumed that 97.5% of the suspect that the health worker screened have a chronic cough (defined as having a cough for two or more weeks). The study anticipated that a majority (88.9%) would give a sputum sample for testing. Whereas 60% of the tested may have TB disease, 77.6% may have the disease (true positive) [Table 1].

When a positive case was identified from PCF, whether true positive or false positive, it was necessary to conduct HCI. The decision model aimed to determine the cost of detecting a single case from both strategies; the terminal nodes are presented as detecting > = 1 claim or 0 instances. Based on the literature, the study assumed that the probability of detecting > = 1 case from HCI is 19% from an actual positive sputum test. A further assumption was that two

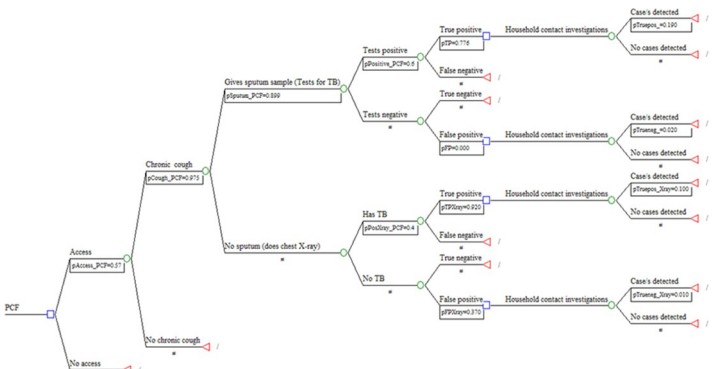

**Fig 1. Decision tree for passive case finding and household contact investigations.**

(2%) from a false positive sputum test, 10% from a true positive from a chest X-ray, and 1% from a false positive chest X-ray [Fig 1].

## Incremental cost-effectiveness ratio

The study used costing in the 2021 US Dollars rate. We computed an incremental cost-effectiveness analysis of PCF+HCI strategies compared to PCF alone per additional case detected. We calculated the incremental cost-effectiveness ratios (ICERs) by dividing the incremental cost by the incremental effectiveness. We compared the ICERs to cost-effectiveness thresholds defined by the country's annual gross domestic product (GDP) [21]. The interventions are cost-effective when ICERs fall within one to three times the annual GDP per capita [30]

## Sensitivity analysis

We performed One-way sensitivity analyses to explore the impact of uncertainty in the base analysis's sensitivity, specificity, and cost estimates. In one-way sensitivity analyses, each selected variable varies one at a time over a range of predefined clinically and economically plausible maximum and minimum values. The sensitivity analysis, the impact of uncertainties

**Table 1. Probabilities estimates used in this decision analytic model.**

| Variables | Variable definitions | Measure (CI) | Sources/References |
|---|---|---|---|
| pAccess_PCF | Probability of accessing care | 0.570 (0.250–1.000) | WHO |
| pCough_PCF | Probability of ≥2 weeks cough having accessed the facility | 0.975 (0.780–1.000) | Uganda TB Program records |
| pSputum_PCF | Probability of having a sputum test | 0.899 (0.750–0.950) | Uganda TB Program records |
| pPositive_PCF | Probability of sputum testing positive | 0.600 (0.200–0.750) | Uganda TB Program records |
| pTP | Probability of true positive | 0.776 (0.610–1.000) | [12] |
| pFP | Probability of true negative | 1.000 (0.883–1.000) | [12] |
| pPosXray_PCF | Probability of positive TB from X-ray | 0.400 (0.300–0.700) | [12] |
| pTPXray | Probability of true positive from X-ray | 0.920 (0.700–0.950) | [12] |
| pFPXray | Probability of true negative from X-ray | 0.630 (0.520–0.990) | [12] |
| pTruepos_ | Probability of detecting ≥1 case from HHCI after true sputum positive | 0.190 (0.060–0.240) | [12] |
| pTrueneg_ | Probability of detecting ≥1 case from HCI after false sputum positive | 0.020 (0.000–1.000) | [12] |
| pTruepos_Xray | Probability of detecting ≥1 case from HCI after true X-ray positive | 0.100 (0.000–1.000) | [12] |
| pTrueneg_Xray | Probability of detecting ≥1 case from HHCI after false X-ray positive | 0.010 (0.000–1.000) | [12] |

on the probabilities, and cost estimates were explored. The probabilities used in the model and the costs are varied one at a time using values in the confidence intervals presented in [Table 7] below. This analysis shows probabilities over extreme ranges of zero to one for cases detected from an accurate positive index or positive chest X-ray index in PCF + HCI due to more significant uncertainty in the base values.

### Ethical approval and consent

We registered the study with the Uganda National Council of Science and Technology (UNCST) (2021–100) and obtained institutional ethical approvals from the Gulu University Research Ethics Committee **GUREC–2021–100 Cost-Effectiveness of adding Tuberculosis Household Contact Investigation on Passive Case Finding strategy in Southwestern Uganda**. The study sought administrative permission from the Lira University Faculty of Health Sciences, Districts, and facilities authority. We obtained approval and written informed consent from all participants to ensure the protection and confidentiality of all study information and their privacy [S7 File]. Furthermore, we excluded children (minors) (< 18 years) from the cost survey because they could not provide accurate data and information on cost estimates.

## Results

### The participant's socio-demographic characteristics of index patients and their households contacts

Twelve health facilities registered 670 index TB cases in contact tracing and unit TB registers from January 2020 to June 2021 in Southwestern Uganda. The 670 TB index patients had 3454 household contacts, with an average household size of five family members (SD: ±3.2.01). The median age of all index TB cases was 38 years (IQR: 0.3–98) and 18 years (IQR: 0.1–107) for the household contacts. The facilities identified more males at 66.1% (443/670) among the index TB cases; however, the study found more females among household contacts at 54.8% (1893/3454). About 32.1% (215/ 670) of the index clients and 4.1% (139/3454) of the household contacts tested HIV positive during the study period [Table 2].

### The health facility's Passive Case Finding (PCF) TB screening and diagnostic cascade

There were 358608 patients reported in the District Health Information System. More females visited facilities at 72.53% (260101/358608) than males. Only 13.6% (48789/358608) of children below 0–14 years came to health facilities across the 12 health facilities from January 2020 to June 2021 [Table 3]. The health facilities screened 80.6% (289140/358608) of patients for TB [Table 3]. The proportion of people screened for TB was high in HIV Clinics at 86.0% (74126/ 86240) and lowest in OPD, 77.3% (155595/201178). The overall presumptive rate was 4.5% (12998/289140). However, more males at 6.9% (5,700/82,624) had the highest presumptive rate than females, and MCH (ANC) clinics had the lowest presumptive rate at 0.3% (182/ 59,419).

Disaggregated by age and sex, 81.2% (251588/309819) of 15+ years and 77.0% (37552/ 48789) of children below 0–14 years received screening for TB. Among the presumptive TB patients diagnosed, males accounted for the highest proportion of those diagnosed with TB at 16.2% (923/5700) [Table 3].

**Table 2. The participant's socio-demographic characteristics of index clients and their household contacts.**

| Study profile | Parameters | Index TB cases (PCF) | Household contacts |
|---|---|---|---|
| Number | Total | N = 670 | N = 3454 |
| | Average (±SD) | | 5 (3.201) |
| | Median (IQR) | | 5 (1–27) |
| Age | Average (±SD) | 36 (21.848) | 23.6 (19.98) |
| | Median (IQR) | 38 (0.3–98) | 18 (0.1–107) |
| Sex | Male, N (%) | 443 (66.1%) | 1561 (45.2%) |
| | Female, N (%) | 227 (33.9%) | 1893 (54.8%) |
| HIV Status | HIV Positive, N (%) | 215 (32.1%) | 139 (4.1%) |
| | HIV Negative, N (%) | 449 (67.0%) | 2792 (80.8%) |
| | Unknown, N (%) | 6 (0.9%) | 523 (15.1%) |

Source: Primary Data from Contact tracing register, Unit TB register (January 2020–June 2021), SD = Standard Deviation, IQR = Interquartile Range, N = number, % = proportion

## The household contact investigation (HCI) screening and TB diagnostic cascade

Facilities screened 3414 contacts from 3454 registered participants, representing 98.8% compared to 80.6% of participants screened from PCF. The study revealed that presumptive TB was one in every six household contacts (3414/531). The proportion of presumptive cases was highest among PLHIV, 25.9% (36/139). The proportion of contacts diagnosed with TB was higher among contacts 0–14 years at 48.7% and menat 39.6%. In addition, the proportion of presumptive diagnosed with TB was highest among PLHIV, 61.1% (22/36) [Table 4].

## TB screening yields passive case finding compared with household contacts

The PCF strategy's overall screening yield was 0.52% (520/100000). The screening yield was four times higher among men, at 1.12% (1200/100000), $P < 0.01$, and lowest in MCH (Antenatal clinics) at 0.02% (20/100000), $P < 0.01$. The HCI strategy showed that the overall screening yield was 5.8% and 11.2 times higher than PCF (0.52%) [Tables 3 and 4]. The Household contact tracing also resulted in a high screening yield among children 0–14 of 6.2%, $P = 0.04$. The screening yield among PLHIV was 15.8% (158/1000) and 2.7 times higher than the overall screening yield of 5.8% in HCI [Table 4].

## The NNS in passive case finding compared with household contact investigation

The overall NNS from HCI was 17 (95% CI: 14–22) and 11 times lower than in the PCF (193) strategy. Notably, the NNS was 38 times higher in MCH (ANC) entry care points than OPD 5402 (95% CI: 5347–6526) Vs. 144 (95% CI: 120–253) for PCF [Table 3]. The study found lower NNS of 6 (95% CI: 2–6) among PLHIV [Table 4].

## The cost of identifying a TB case using different case-finding strategies

The total costs for PCF and HCI were US$ 266,093.27 and US$ 61,005.07, respectively, with unit costs of US$ 204.22 for PCF and US$ 315.07 for HCI. The proportion of total program

**Table 3. Passive case finding TB screening and diagnostic cascade, yield, and NNS disaggregated by age, sex, and entry care points.**

| Characteristics | Patients seen (N) | Screened N (%) | Presumptive TB N (%) | Presumptive TB Investigated N (%) | Presumptive TB Diagnosed N (%) | Screening Yield (%) | NNS (95% CI) | P—Value† |
|---|---|---|---|---|---|---|---|---|
| **Overall Number** | **358608** | **289140 (80.6)** | **12998 (4.5)** | **11049 (85)** | **1496 (11.5)** | **0.52** | **193 (186–294)** | |
| **Sex** | | | | | | | | |
| **Female** | 260101 | 206516 (79.4) | 7298 (3.5) | 6349 (87) | 573 (7.9) | 0.28 | 360 (300–508) | < 0.01 |
| **Male** | 98507 | 82624 (83.9) | 5700 (6.9) | 5073 (89) | 923 (16.2) | 1.12 | 90 (53–161) | |
| **Age** | | | | | | | | |
| **0–14** | 48789 | 37552 (77.0) | 1871 (5.0) | 1684 (99) | 288 (15.4) | 0.77 | 130 (115–223) | < 0.01 |
| **15+** | 309819 | 251588 (81.2) | 11127 (4.4) | 9347 (84) | 1208 (10.9) | 0.48 | 208 (196–316) | |
| **Entry care Points** | | | | | | | | |
| **OPD** | 201178 | 155595 (77.3) | 8566 (5.5) | 7110 (83) | 1081 (12.6) | 0.69 | 144 (120–253) | < 0.01 |
| **HIV Clinic** | 86240 | 74126 (86.0) | 4250 (5.7) | 4080 (96) | 404 (9.5) | 0.55 | 183 (152–276) | |
| **MCH (ANC)** | 71190 | 59419 (83.5) | 182 (0.3) | 66 (36) | 11 (6.0) | 0.02 | 5402 (5347–6526) | |
| **Health facility level** | | | | | | | | |
| **Health Center II** | 17820 | 1747 (96.2) | 612 (3.4) | 612 (100) | 113 (18.5) | 0.66 | 152 (104–212) | < 0.01 |
| **Health Center III** | 94845 | 78071 (82.3) | 2402 (2.5) | 2138 (89) | 381 (15.9) | 0.49 | 205 (195–303) | |
| **Health Center IV** | 184451 | 145756 (79.0) | 8800 (4.8) | 6952 (79) | 865 (9.8) | 0.59 | 169 (159–267) | |
| **Hospital** | 61492 | 48166 (78.3) | 1184 (1.9) | 853 (72) | 137 (11.6) | 0.28 | 352 (347–356) | |

Source: DHIS 2 (OPD Register, Ante Retroviral Register, Unit TB, ANC Register, Presumptive and Unit TB laboratory registers). NNS: Number Needed to Screen, OPD: Outpatient Department, MCH: Maternal & Child Health, ANC: Antenatal Clinic,

† Likelihood Ratio P-values obtained from Poisson regression for Rates (that is, number of patients diagnosed with TB over the total number screened over the One and Half years studied)

costs was higher in HCI at 84.87%, and direct medical costs were higher in PCF at 91.54% (US $ 238,414.89)of the total healthcare costs [Table 5].

The overall patient and caregiver costs were five times in PCF than in HCI (US$ 26.37 Vs. US$ 5.42) [S1 File]. Meals accounted for the highest Cost of US$ 6.10 in direct patient cost; however, the average time lost accounted for the most Cost at US$ 7.8 under PCF indirect patient costs [Table 5].

Under PCF, 89.09% of the cost went to performing MTB/RIF/Ultra GeneXpert Tests, followed by staff salaries at 7.06%, while 2.45% when to Consumables and Supplies. The nominal costs were printing, copying, and office supplies at 1.37% and phone communication at 0.03%. HCI costs were allowance and transport at 38.10%; Training at 21.27%; MTB RIF/Ultra GeneXpert at 13.14%; supervision by the District TB and Leprosy Supervisor at 8.23%, printing, copying, and office supplies at 8.04%. Staff salaries and supplies consumables accounted for 9.32% and 1.69%, respectively [Table 5].

**Table 4. Description of contacts screened through household contact investigations.**

| Characteristics | Contacts seen (N) | Screened N (%) | Presumptive TB N (%) | Presumptive TB Investigated N (%) | Presumptive TB Diagnosed N (%) | Screening Yield (%) | NNS (95% CI) | P—Value† |
|---|---|---|---|---|---|---|---|---|
| **Overall Number** | **3454** | **3414 (98.8)** | **531 (15.6)** | **381 (71.8)** | **197 (37.1)** | **5.8** | **17 (14–22)** | |
| **Sex** | | | | | | | | |
| **Female** | 1894 | 1870 (98.7) | 296 (15.8) | 200 (67.6) | 104 (35.1) | 5.6 | 18 (14–22) | < 0.01 |
| **Male** | 1560 | 1543 (98.9) | 235 (15.2) | 181 (77.0) | 93 (39.6) | 6.0 | 17 (13–21) | |
| **Age** | | | | | | | | |
| **0–14** | 1477 | 1460 (98.8) | 187 (12.8) | 187 (100) | 91 (48.7) | 6.2 | 16 (12–20) | 0.04 |
| **15+** | 1977 | 1954 (98.8) | 344 (17.6) | 250 (72.7) | 106 (30.8) | 5.4 | 18 (15–23) | |
| **HIV Status** | | | | | | | | |
| **Positive** | 139 | 139 (100) | 36 (25.9) | 30 (83.3) | 22 (61.1) | 15.8 | 6 (2–10) | 0.03 |
| **Negative** | 2792 | 2776 (99.4) | 425 (15.3) | 297 (69.9) | 147 (34.6) | 5.3 | 19 (15–23) | |
| **Unknown** | 523 | 499 (95.4) | 70 (14.0) | 54 (77.1) | 28 (40.0) | 5.6 | 18 (15–23) | |

Source: Primary Data (Contact tracing, Unit TB register), NNS: Number Needed to screen,

†Likelihood Ratio P-values obtained from Poisson regression for Rates (that is, Number of TB patients diagnosed with TB over the total number screened for TB over the One and Half years studied

**Table 5. Summary of cost (in 2021US$) estimates associated with TB detection from [S1 File].**

| Cost category | Cost, $ | Range (+/- 50%) | Source of data |
|---|---|---|---|
| **Program costs 1** | | | |
| PCF | 15.04 | 7.52–22.56 | Uganda TB program records 2020–2021 |
| HCI | 262.79 | 131.39–394.18 | Program activity Budget |
| PCF + HCI | 277.83 | 138.92–416.75 | Uganda TB program records 2020–2021, Program activity Budget |
| **Direct Medical 2** | | | |
| PCF | 162.81 | 81.41–244.22 | Uganda TB program records 2020–2021 |
| HCI | 46.85 | 23.43–70.28 | Program activity Budget |
| PCF + HCI | 209.66 | 104.83–314.50 | Uganda TB program records 2020–2021, Program activity Budget |
| **Total Patient &Caregiver Costs 3** | | | |
| PCF | 26.37 | 13.18–39.55 | Uganda TB program records 2020–2021 from Cost Survey. |
| HCI | 5.42 | 2.71–8.14 | Uganda TB program records 2020–2021, Program activity Budget. |
| PCF + HCI | 31.79 | 15.90–47.69 | Uganda TB program records 2020–2021, Program activity Budget. |
| **Total per Patient Costs 4** | | | |
| PCF | 204.22 | 102.11–306.33 | Uganda TB program records 2020–2021 form cost Survey |
| HCI | 315.07 | 157.53–472.60 | Uganda TB program records 2020–2021, Program activity Budget |

1. Program costs include administration, transport, communication & health personnel

2. Direct medical costs include MTB RIF/Ultra GeneXpert & Supplies & Consumables

3. Total patient and caregiver costs include direct (transportation& meals) and Indirect costs (productivity/wages lost) 4. Estimated total per-patient costs are a summation of program, direct medical, and total patient-caregiver costs estimated

**Table 6. Incremental cost-effectiveness analysis with PCF as the baseline using the health provider perspective.**

| Strategy | Total cost (US$) [a] | Incremental Cost | Total effects[a] | Incremental effects[b] | Total cost/ total effect (ACER) | ICER[b] (Cost per additional case detected) |
|---|---|---|---|---|---|---|
| PCF alone | 45,392 | - | 253 | - | 179.42 | - |
| PCF + HHCI | 214,448 | 169,056 | 300 | 47 | 714.83 | 3,596.94* |

[a]. The study rounded effects to the nearest whole number per 1000 persons screened in the target population.

[b]. ICER- incremental cost-effectiveness ratio/*calculations were based on five significant digits to increase precision and minimize rounding errors.

## Model findings and incremental cost-effectiveness result

The expected value for TB cases detected was 0.253, and the expected cost was $453.92. The decision of an optimal path was based on the model path that generates the highest expected value of cases detected [S2 File].

## Cost-effectiveness analysis (CEA)

The expected number of TB cases per 1000 persons screened in PCF alone was 253 at an average cost of $45,392. PCF + HCI was 300 at $214,448 [S2 File]. The estimated costs and effects were calculated from the baseline average and incremental cost-effectiveness analysis comparing PCF alone to PCF + HCI.

## Average cost per true TB case detected

The average cost-effectiveness ratios (ACERs) demonstrate the cost per case detected in PCF and PCF + HCI compared to doing nothing at zero cost. The average price of detecting an actual TB case in PCF alone was $179.42, and $714.83 in PCF + HCI [Table 6].

## Incremental cost-effectiveness of PCF vs. incorporating HCI

The incremental effect of adding HHCI was 47 cases, costing $3,596.94 per case detected [Table 6].

## The sensitivity analysis

The sensitivity analysis indicated that the model was most sensitive when changes were made to the probability of detecting > = 1 case from Household contact investigation and after identifying a true positive case from passive case finding as an index case. When the likelihood of true positive increases to 0.24, the ICER decreases to 14 compared to 16.91 at base PCF. This indicates that incorporating HCI into PCF may become cost-effective as the true positivity rate increases [Fig 2]. The rest of the model parameters did not have a meaningful impact on the conclusion of the analysis [Table 7].

## Discussion

The study compared the cost and the incremental cost-effectiveness ratio adjusted to US$ 2021, the TB yield, and the number needed to screen from passive case finding (PCF) and household contact investigation (HCI). The incremental cost-effectiveness ratio (ICER) represented the cost to detect an additional TB case and decision threshold based on Uganda's GDP per capita ($858.1 - $2,574.3) [31]. In this study, we observed that PCF had lower unit costs

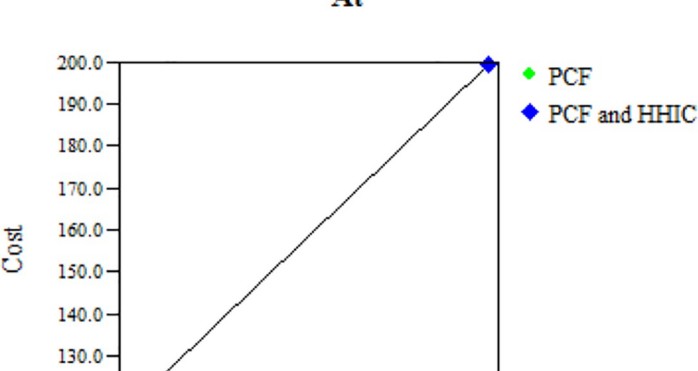

**Fig 2. Cost-effectiveness analysis from sensitivity test.**

than HCI (US$ 204.22 vs. US$ 315.07). The finding was expected since PCF is the least labor-intensive strategy. The average cost for detecting a unit TB case in this study compared reasonably with those obtained from similar settings. For example, the median cost was $132.00 (US $60.00–1626.00) per TB case detected, as reported from eight active case-finding projects in high TB burden countries, including Uganda. The projects were conducted under the Fund for Innovative DOTS Expansion through Local Initiatives Stop TB [32].

In contrast, the unit cost of carrying out both strategies was higher for PCF and lowered for HCI in this study than the Kakinda and Matovu study, which reported US$ **120.6** for PCF and US$ **877.57** for HCI [11]. The most probable reason is that the Kakinda and Matovu study had a shorter duration (i.e., six months) than this study, where the evaluation lasted one and a half

**Table 7. One-way sensitivity analysis for cost-effectiveness of TB case finding strategies (PCF and HCI) varying probabilities.**

| | | ICER (US$/ TB case detected) | | For Low Value | | For High Value | |
|---|---|---|---|---|---|---|---|
| **Strategies compared** | | PCF + HCI vs. PCF | | PCF | PCF + HCI | PCF | PCF + HCI |
| **Base ICER**[a] | | 16.91 | | 453.92 | 2,144.48 | 453.92 | 2,144.48 |
| **Variables** | Measure (CI)[b] | low | high | | | | |
| **pAccess_PCF** | 0.570 (0.250–1.000) | 346.01 | 93.18 | 476.71/ 0.111 | 2,206.78/ 0.116 | 446.26/ 0.444 | 2,123.55/ 0.462 |
| **pCough_PCF** | 0.975 (0.780–1.000) | 213.93 | 1688.47 | 465.96/ 0.203 | 2,177.41/0.211 | 452.71/0.27 | 2,141.18/ 0.26 |
| **pSputum_PCF** | 0.899 (0.750–0.950) | 106.60 | 81.09 | 464.52/0.245 | 2,383.34/0.227 | 450.44/0.256 | 2072.34/0.276 |
| **pPositive_PCF** | 0.600(0.200–0.750) | 51.13 | 7765.36 | 1,170.78/0.098 | 4,443.21/0.034 | 369.15/0.311 | 1,891.16/0.115 |
| **pTruepos_** | 0.190 (0.060–0.240) | 28.33 | 14.00[c] | 453.92/0.253 | 6,176.18/0.051 | 453.92/0.253 | 1,714.11/0.343 |

[a] ICER = Incremental Cost-Effectiveness Ration

[b] Ranges obtained from published literature, expert opinion, or full ranges used

[c] PCF+HHCI becomes a cost-effective strategy at ICER $14.00

pAccess_PCF (Probability of accessing care in PCF), pCough_PCF (Probability of chronic cough given access), pSputum_PCF (Probability of sputum production & TB test given chronic cough), pPositive_PCF (Probability of positive case in PCF), pTruepos_ (Probability of true positive in combined sensitivity of GeneXpert Test)

years. The study duration could have helped the study team diagnose more TB cases, reducing the cost per case in household contact investigation. Further, the low cost of conducting HCI in this study compared to the Kakinda and Matovu study could be due to the geographical settings (facilities and region) while performing HCI. It is costlier to carry out HCI in Regional Hospitals than in lower facilities, as in Kakinda and Matovu's study, since clients diagnosed come from various regional districts and facilities. Moreover, this would increase the cost of transportation and allowances for healthcare working across districts. Following TB contacts by the nearby districts and facilities within a TB client's home would reduce the high cost of carrying out household contact investigations by the regional hospital.

The overall patient and caregiver costs for direct and indirect out-of-pocket were five times in PCF than in HCI (US$ 26.37 Vs. US$ 5.42). Our study agrees with a similar patient cost survey conducted in Uganda that reported the overall patient and caregiver costs for direct and indirect out-of-pocket was six times in PCF than in HCI (US$ 28.88 Vs. US$ 4.76) [12]. Food or meals and the average time lost accounted for these highest costs under PCF at US$ 6.10 and US$ 7.8, respectively. Even though the HCI was more expensive per patient diagnosed, it had less out-of-pocket expenditure with fewer health facility visits than those diagnosed from the standard passive system. Notably, most studies on the economic burden of TB have demonstrated that patient costs primarily arise from direct out-of-pocket expenses on transportation, hospitalization, and meals [21] and, to a varying extent, from indirect costs from patients 'time lost from work or death. Moreover, passive case finding result in high-income losses before diagnosis and treatment due to health system delays [21]. People incur substantial direct and indirect costs to access care for TB in settings where programs provide free TB services [33].

This study's ICER for HCI to PCF costs US$ 3,596.94 per case detected by providers and societal perspectives. The ICER for incorporating HCI into PCF falls outside the set decision threshold of 1 to 3 times Uganda's GDP per capita ($858.1 - $2,574.3) [21]. From a health provider and social standpoint, combining PCF with HCI proved ineffective in detecting TB cases rather than HCI as a strategy. Our findings cost more than those reported by [12], who found that it cost $443.62 to diagnose one TB case. The difference could be due to the analysis and mode of implementation costs incurred by the District Tuberculosis and Leprosy focal person during supervision and monitoring and higher training costs. The other reason could be the prevalence of TB and the geographical context, rural versus urban settings. The prevalence of TB tends to be higher in urban slum settings than in rural areas [34]. It means we need to spend more money to diagnose TB cases in rural areas with a lower prevalence in the population. Policy decision-makers should view cost-effectiveness results considering other context-specific factors such as disease burden, the patient mix in the target population, and ethics and equity.

The sensitivity analysis indicated that the model was most sensitive when changes were made to the probability of detecting $> = 1$ case from Household contact investigations after identifying a confirmed positive case from a passive case finding as an index case. When the likelihood of true positive increases to 0.24, the ICER decreases to 14 compared to 16.91 at base PCF. This indicates that incorporating HCI into PCF may become cost-effective as the true positivity rate increases. It is not surprising because the risk of household contact becoming a TB case is primarily driven by the infectiousness of the index case [35, 36]. It is critical to note that the cost and effectiveness of an HCI strategy would be highly dependent on a well-functioning PCF program and its ability to detect index cases and timely follow-up of household contacts. In Uganda and much of Africa, the lack of well-organized health systems, a shortage of healthcare personnel, and limited adequate health resources pose practical barriers to implementing household contact investigations [12]. However, in the real world, adding

HCI to PCF is an optimal strategy to prioritize and regularly evaluate for efficiency according to changes in disease epidemiology and other population dynamics.

Despite the higher cost, household contact investigation could be an effective strategy. TB clients will likely be detected early in contact investigation, decreasing morbidity and mortality. Similarly, the negative consequences for low TB case detection in the PCF are not factored in this analysis because of the short time horizon. In the PCF strategy, people with TB are treated or only treated after a long delay, contributing to the sustained burden, morbidity, and mortality. Ignoring these consequences in the PCF costs could underestimate HCI's cost-effectiveness.

Additionally, a household contact investigation might reduce the patient's stigma and discrimination since the household better understands the disease. Nevertheless, it may be helpful to integrate household contact investigation with TB preventive therapy (TPT) provision, which has significant implications for TB burden (cases averted) and TB/HIV epidemic control in the medium and long term. The costs for TPT provision and HIV Counseling and Testing services (HTS) can be subsidized through HCI. Our study did not consider the medium and long-term benefits of TPT provision.

The yield from household contact tracing was 11 times higher than PCF in this study (5.8% vs. 0.52%), suggesting it is more effective for diagnosing Tuberculosis in our setting. It aligns with current knowledge and other studies that reported high screening yield among household contacts [37, 38]. In contrast, the result revealed that the Southwestern TB control program in three Districts had a higher yield than Uganda's annual household TB screening yields of 3.3% and 0.3% for HCI and PCF, respectively [16]. Differences in TB burden and effectiveness of TB program coverage in the Southwestern region could partly explain the disparities. Our study sampled 12 facilities in the Southwestern region where TB linkage facilitators overcome previously documented causes of non-completion of the TB screening cascade. These include long waiting times at the health facilities and a lack of time or transport money for household contacts to visit the health facilities [39–41]. In addition, the household contact tracing yield is comparable to a study conducted in 12 hospitals in Uganda, which reported a TB yield of 5.29% among contacts screened [11].

The long-term effect of household contact investigation may depend on increased numbers notified and its ability to find TB patients before they transmit TB. It occurs at a stage when the symptoms are still limited and have not prompted the individual to seek care. The household as an entry point for contact tracing and screening could result in about 40% of people with infectious pulmonary TB not being missed, as per the 2014/2015 Uganda TB prevalence survey findings. The significantly higher Yield of active TB among contacts may suggest such potential. However, the full potential effect of contact tracing on TB case finding and transmission can only be assessed through randomized controlled trials or step wedge design [41].

The screening yield was different among females and males in both HCI at 5.5% (55/1000) in females, and males, 6.0% (60/1000) $P<0.01$, and in PCF at 0.28 (280/100000) in females and males at 1.12 (1200/100000) $P<0.01$. The higher screening yield among men in both strategies is consistent with findings from several TB prevalence surveys and studies that report a higher prevalence of TB among men [42, 43]. It could be because men are sometimes more exposed to predisposed factors like alcohol and smoking. Studies show that generally, men smoke more than women [44]. The results of this study were like findings from a study in Nigeria in which the prevalence rate observed was high among males. The study indicated that men experience more exposure to the world than women (mainly in rural areas), including social relations with others and higher risk of being exposed to TB infected individuals, thus getting the possibility of becoming infected with TB [45]. 'It calls for further research on the high prevalence of TB in the male gender.

The household contact investigation is an entry point for childhood TB care and people living with HIV. Several studies documented this as an opportunity to timely evaluate the child and HIV co-infected contacts in to treat active disease and latent TB infections [46, 47]. The study finding revealed a high screening yield among children 0–14 of 6.2% in Household contact investigation than in passive case findings (0.52%). Similarly, household contact tracing yields among those living with HIV were 15.8% compared with 0.55% in passive case findings. The findings are consistence with recently reported other studies in similar settings showing a pooled yield of active contact screening at 3.2% (2.9–3.4%) in the general population and the pooled average of 9.0% for people living with HIV (PLHIV) and 6.3% for children [48]. A study in rural Malawi compared the yield of TB disease from passive and active case finding in household contacts. The results showed that TB cases yielded nine times more when using active case finding in HIV-co-infected contacts [49]. Our study actively emphasizes the need for program managers to improve case findings among children and PLHIV by promoting contact tracing among childhood and PLHIV contacts. Contact investigation may detect TB and HIV co-infection more effectively than passive case findings.

On the other hand, contact investigation may be less effective in the absence of well-developed public health infrastructure to follow index cases at their homes. It is plausible that contact investigation would significantly contribute to earlier TB case detection. The additional number of cases detected is less specific, and it's cost-effectiveness warrants evaluation to inform policy decision-making [12]. In conclusion, combining contact investigation with other case-finding approaches would likely increase overall case notification [50].

In this study, the overall NNS from PCF was 193 (95% CI: 186–294) and 17 (95% CI: 14–22) in the HCI strategy. Importantly an NNS of 100 would be a priority intervention for TB Case findings [51]. In practice, at a prevalence of 200 per 100000 population, the NNS is at 500 and will be higher when the screening accuracy is suboptimal [52]. The prevalence of undetected TB in the population is often less than 200/100000, even in high-burden TB countries. Therefore, the WHO does not recommend screening the general population since it is not usually cost-effective [7]. The recommendation confirms that household contact investigation, a targeted active case finding, would be a cost-effective intervention. In addition, the result from this study reported a higher NNS of 5402 ((95% CI: 5347–6526) in Maternal Child Health (MCH). The findings are consistent with a similar study conducted in Kampala that reported a higher NNS of 8410 in the MCH [42]. Higher NNS in MCH could point to the quality of TB screening, which calls for implementers and managers in MCH to strengthen screening quality in this area. However, the number-needed-to screen can vary depending on the screening accuracy and the diagnostic algorithms, the prevalence of TB and HIV in the target population, and the TB control program functionality [22].

It is important to note that the NNS is a rough indicator of cost-effectiveness and effort. Comparing the NNS of risk groups provides relative cost-effectiveness. Moreover, the assumption is that the cost of diagnosis and treatment and the benefits of early treatment are the same for all risk groups (World Health Organization). This assumption is, however, rarely valid in the real world. For example, an NNS of 50 for contact investigation might mean visiting 12 houses over two days. In contrast, an NNS of 150 might benefit from verbal screening in a slum area where many people can be verbally screened and diagnosed in 4 hours. The energy and cost of screening are higher in the first example, even though they lower NNS.

## The study's strengths and limitations

The evaluation had several strengths. The data from this evaluation was collected retrospectively as part of a large ongoing TB implementation project and thus fully reflects

programmatic conditions in low-resource, high-burden settings. The study could be very insightful for progressive policy and research studies. Another strength of this study was the TB screening cascade analysis which would contribute to a growing body of literature on TB case finding that seeks to account for how patients move through the different steps of the screening, diagnostic, and treatment cascade to improve case finding in routine healthcare settings.

A significant strength of this study is that it was a multi-center study. The researcher used probability and cost estimates primarily drawn from actual data in the Uganda National TB Program and primary studies conducted in Uganda. Therefore, the model assumptions are close enough to the actual world situation.

Furthermore, the costs borne by patients were evaluated directly from a cost survey of patients receiving TB services from the PCF system in Southwestern Uganda. It is an addition to the existing cost literature. Moreover, the researcher analyzed the widely practiced TB case detection strategies from the societal perspective as recommended by the Panel on Cost-effectiveness in Health and Medicine [22].

The mathematical modeling work has consistently highlighted the importance of case detection and the economic benefit of ACF (HCI) under certain conditions [53, 54]. It is impossible to compare this study's results and findings from mathematical modeling studies because of differences between the static and dynamic model capabilities and outcome measures. None of the previous models considered household contact investigations among the alternatives for case detection.

The findings in this economic study were subject to some limitations. First, the data for this analysis was collected by healthcare workers involved in routine healthcare delivery, resulting in missing data. However, the proportion of missing data for each variable was relatively small. It did not significantly affect the integrity of the analysis.

Second, using a Decision Tree Model to estimate the cost and effectiveness of the strategies as measured by the number of contact persons screened and actual TB cases detected is a short-term benefit. This limitation could lead to underestimating the long-term health benefits of early detection, particularly for the HCI strategy. Mathematical modeling would be required to overcome this limitation. The study findings may not be non-generalizable to countries where the same sector simultaneously undertakes HCI and PCF strategies. Studies have shown that HCI can identify people with TB earlier and latent TB for prevention and potentially reduce TB transmission [54, 55]. The static model is not well suited to include the effects of TB transmission. However, the conclusion that HCI is not cost-effective could be conservative since the incorporation of reduced transmission was not considered during the analysis.

Third, the model does not explicitly account for the prevalence of TB disease in the general population; this limits the evaluation of the effect of varying prevalence levels on the model results. The results assumed Uganda's current high TB prevalence and other African disease burden settings.

Finally, cost-effectiveness studies are prone to selection or publication bias because of heavy reliance on the published literature for model parameter estimates and the ranges of values used in sensitivity analyses. However, the study obtained estimates from actual data in Uganda, which minimized these biases.

## Conclusion

Under our baseline assumptions and the specific implementation of adding PCF to HCI compared to PCF alone prove to be not cost-effective, rather than HCI as a strategy. However, incorporating HCI into PCF may become cost-effective as the true positivity rate increases.

HCI effectively identifies children and PLHIV with TB and should be prioritized. Meanwhile, the Passive case-finding strategy effectively finds men with TB and costs lower than household contact investigation.

## Recommendations

**Key recommendations.**

1. Given the high yield and lower number needed to screen the population, adding HCI to PCF is effective in early TB case identification and reduces community TB case transmission. However, PCF remains the ideal and more cost-effective strategy for low-resource countries like Uganda.

2. This study actively emphasizes the need for program managers to improve case findings among children 0–14 years and PLHIV by promoting household contact investigation among childhood and PLHIV contacts.

3. The manufacturers should consider subsidizing the Cost of GeneXpert cartridges to offset the high cost of diagnosing TB under both strategies using GeneXpert.

## Areas for future research

1. Research may need to investigate further why there are wide disparities in gender diagnosis of TB cases in the Southwestern region. It may be able to guide the clinicians and program implementers to target the risk factors for primary TB prevention measures among men.

2. Given the paucity of cost-effectiveness evidence, we need studies to examine the lifetime health benefits and costs that accrue from HCI in various local African contexts.

3. Higher NNS in Maternal Newborn Health (MNH) and low screening yield could point to the sensitivity and specificity of screening tools and the quality of TB screening. It calls for implementers and managers in MNH to strengthen the quality of TB screening in this area.

## Supporting information

**S1 File. Cost of identifying a TB case using different case finding strategies (PCF &HHCI).**
(DOCX)

**S2 File. Expected values obtained from the decision model.**
(TIF)

**S3 File. Patient cost survey questionnaire.**
(PDF)

**S4 File. Costing and evaluation data.**
(XLSX)

**S5 File. Healthcare provider questionnaire.**
(PDF)

**S6 File. Data extraction tools.**
(PDF)

**S7 File. Ethical clearance and approvals.**
(PDF)

**S8 File.**
(ZIP)

## Acknowledgments

The authors thank the District TB, Leprosy supervisors (Catherine Amutuheire, Nelson Mandela and Ochom Robert) and healthcare facility staff who participated in the study. We also thank the District Health Officers of Sheema, Ntungamo, and Rwampara, that granted permission to conduct the research at the health facilities. Special thanks go to the pivotal research assistants notably Marion Nahabwe in interviewing and collecting the data.

## Author Contributions

**Conceptualization:** Dickens Odongo.

**Data curation:** Dickens Odongo.

**Formal analysis:** Dickens Odongo.

**Investigation:** Dickens Odongo.

**Methodology:** Dickens Odongo.

**Project administration:** Dickens Odongo.

**Resources:** Dickens Odongo.

**Software:** Dickens Odongo.

**Supervision:** Dickens Odongo, Bernard Omech, Alfred Acanga.

**Validation:** Dickens Odongo, Bernard Omech, Alfred Acanga.

**Visualization:** Dickens Odongo.

**Writing – original draft:** Dickens Odongo.

**Writing – review & editing:** Dickens Odongo, Bernard Omech, Alfred Acanga.

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
