## [Decision Letter · Decision Letter 0]

11 May 2023

PONE-D-23-05092Cost-Effectiveness Analysis of Adding Tuberculosis Household Contact Investigation on Passive Case-Finding Strategy in Southwestern Uganda.PLOS ONE

Dear Dr. Odongo,

Thank you for submitting your manuscript to PLOS ONE. After careful consideration, we feel that it has merit but does not fully meet PLOS ONE’s publication criteria as it currently stands. Therefore, we invite you to submit a revised version of the manuscript that addresses the points raised during the review process.

We look forward to receiving your revised manuscript.

Kind regards,

Hamufare Dumisani Dumisani Mugauri, Ph.D. Public Health

Academic Editor

PLOS ONE

Journal Requirements:

Reviewers' comments:

Reviewer's Responses to Questions

**Comments to the Author**

1. Is the manuscript technically sound, and do the data support the conclusions?

Reviewer #1: Partly

Reviewer #2: Yes

2. Has the statistical analysis been performed appropriately and rigorously? 

Reviewer #1: Yes

Reviewer #2: Yes

3. Have the authors made all data underlying the findings in their manuscript fully available?

Reviewer #1: Yes

Reviewer #2: Yes

4. Is the manuscript presented in an intelligible fashion and written in standard English?

Reviewer #1: Yes

Reviewer #2: Yes

5. Review Comments to the Author

Reviewer #1: Manuscript does not meet style/formatting guidelines of the journal. Pages are landscape and do not contain line numbers.

“It entails screening for Tuberculosis infection, with or without latent TB infection” confusing wording. I believe the guidelines say that contacts do not need confirmation of a TB infection to receive TPT.

The analysis of is well described and clearly presented. However, the finding needs to be more nuanced. The authors note that reductions in TB transmission (and thus future TB burden) are not accounted for in their costing analysis; these are tangible benefits which are being disregarded. This results in underestimation of effectiveness, which is acknowledged as a limitation. At the same time, the negative consequences for low TB case detection in the PCF arm are not factored into the analysis. People with TB who are not treated, or only treated after a long delay, contribute to sustained burden, morbidity and mortality. Ignoring these consequences in the PCF costs again under estimates the cost effectiveness of HCI. This is not acknowledged anywhere in the manuscript. Thus, it feels very difficult for the review to accept “adding [HCI] to the [PCF] in the Southwestern Uganda Tuberculosis control program was not cost-effective”. In addition, the discussion makes no comment on the lack of TPT provision, which should have major implications for TB burden (cases averted) and HIV epidemic control in the medium and long term. Costs for TPT provision can be subsidized through HCI. They also do not comment on the entry point for contact investigation – TB symptoms – which could result in about 40% of people with infectious pulmonary TB being missed (according to Uganda prevalence survey findings). Given that HCI is one of the most basic package of services in TB care, I recommend the authors word their findings more carefully. Something like, this specific implementation of PCF+HCI proved to be not cost-effective, rather than HCI as a strategy. There is some context for the analysis’s limitations in the discussion, but strongly worded sentences are included elsewhere which can overstate the findings.

Reviewer #2: While the abstract is clearly presented by the authors, they miss out on key pieces of information required for a cost effectiveness study. For example in the abstract there is no clear mention if any discounting rates were applied in this study, to allow readers to understand the implications of such decisions in the analysis. it was also critical for the authors to clearly state how the sensitivity analysis was used to address uncertainty.

The methods and results sections are comprehensive and the authors should be applauded for sharing the key ingredients of data that went into this analysis. However, it is critical for the authors to mention if this was a synthesis based evaluation of already existing data and to which extent this data was used. This can be mentioned in the abstract

6. PLOS authors have the option to publish the peer review history of their article (what does this mean?). If published, this will include your full peer review and any attached files.

Reviewer #1: No

Reviewer #2: **Yes: **Howard Nyika

---

## [Author Response · Author response to Decision Letter 0]

29 Jun 2023

A rebuttal letter that responds to each point raised by the academic editor and reviewer(s).

1. Please ensure that your manuscript meets PLOS ONE's style requirements, including those for file naming. We have taken care of this throughout the manuscript.

We have made changes to the cover letter, repository information of my data can obtain after seeking a written permission from the Assistance Commissioner of the Uganda National Tuberculosis and Leprosy program, Kampala Uganda. 

Reviewers' comments:

Reviewer's Responses to Questions and Responses from the Author

Comments to the Author

1. Is the manuscript technically sound, and do the data support the conclusions?

Reviewer #1: Partly

Reviewer #2: Yes

2. Has the statistical analysis been performed appropriately and rigorously?

Reviewer #1: Yes

Reviewer #2: Yes

3. Have the authors made all data underlying the findings in their manuscript fully available?

The PLOS Data policy requires authors to make all data underlying the findings described in their manuscript fully available without restriction, with rare exception (please refer to the Data Availability Statement in the manuscript PDF file). The data should be provided as part of the manuscript or its supporting information or deposited to a public repository. For example, in addition to summary statistics, the data points behind means, medians and variance measures should be available. If there are restrictions on publicly sharing data—e.g., participant privacy or use of data from a third party—those must be specified.

Reviewer #1: Yes

Reviewer #2: Yes

4. Is the manuscript presented in an intelligible fashion and written in standard English?

Reviewer #1: Yes

Reviewer #2: Yes

5. Review Comments to the Author

Reviewer #1: Manuscript does not meet style/formatting guidelines of the journal. Pages are landscape and do not contain line numbers. We have corrected the style/formatting guidelines of the manuscript; pages are in portrait and has line numbers. 

“It entails screening for Tuberculosis infection, with or without latent TB infection” confusing wording. I believe the guidelines say that contacts do not need confirmation of a TB infection to receive TPT. It entails screening for Tuberculosis disease, with or without latent TB infection.” corrected.

The analysis of is well described and clearly presented. However, the finding needs to be more nuanced. The authors note that reductions in TB transmission (and thus future TB burden) are not accounted for in their costing analysis; these are tangible benefits which are being disregarded. This results in underestimation of effectiveness, which is acknowledged as a limitation. At the same time, the negative consequences for low TB case detection in the PCF arm are not factored into the analysis. People with TB who are not treated, or only treated after a long delay, contribute to sustained burden, morbidity, and mortality. Ignoring these consequences in the PCF costs again underestimates the cost effectiveness of HCI. This is not acknowledged anywhere in the manuscript. Thus, it feels very difficult for the review to accept “adding [HCI] to the [PCF] in the Southwestern Uganda Tuberculosis control program was not cost-effective”. We have included the comments in the discussion section (page 22) and acknowledge them in the limitation. We did not factor some of the long-term effects of HCI (Death Averted, Life saved, reduction of burden of TB, transmission) because of short analytical time horizon of this study. in addition, the discussion makes no comment on the lack of TPT provision, which should have major implications for TB burden (cases averted) and HIV epidemic control in the medium and long term. Costs for TPT provision can be subsidized through HCI. They also do not comment on the entry point for contact investigation – TB symptoms – which could result in about 40% of people with infectious pulmonary TB being missed (according to Uganda prevalence survey findings). We appreciated these comments and has been included in our manuscript, Discussion section page 22. Given that HCI is one of the most basic packages of services in TB care, I recommend the authors word their findings more carefully. Something like, this specific implementation of PCF+HCI proved to be not cost-effective, rather than HCI as a strategy. We have taken this recommendation seriously and included in the Conclusion section of the manuscript and abstract. There is some context for the analysis’s limitations in the discussion, but strongly worded sentences are included elsewhere which can overstate the findings.

Reviewer #2: While the abstract is clearly presented by the authors, they miss out on key pieces of information required for a cost effectiveness study. For example, in the abstract there is no clear mention if any discounting rates were applied in this study, to allow readers to understand the implications of such decisions in the analysis. We did not apply a discount rate because of the short analytic time horizon. This was mentioned in the abstract page 2 and methodology section page 7 of the manuscript. it was also critical for the authors to clearly state how the sensitivity analysis was used to address uncertainty. It was stated in the result section of page 12 and Discussion section of page 21 of the manuscript. 

The methods and results sections are comprehensive, and the authors should be applauded for sharing the key ingredients of data that went into this analysis. However, it is critical for the authors to mention if this was a synthesis-based evaluation of already existing data and to which extent this data was used. This can be mentioned in the abstract. It was a synthesis-based evaluation of already existing data and data was extracted from the national system, as well as from the available primary tools at the facility. We mentioned it in the abstract and methodology section.

---

## [Editor Report · Decision Letter 1]

5 Jul 2023

Cost-Effectiveness Analysis of Adding Tuberculosis Household Contact Investigation on Passive Case-Finding Strategy in Southwestern Uganda.

PONE-D-23-05092R1

Dear Dr. Odongo,

We’re pleased to inform you that your manuscript has been judged scientifically suitable for publication and will be formally accepted for publication once it meets all outstanding technical requirements.

Kind regards,

Hamufare Dumisani Dumisani Mugauri, Ph.D. Public Health

Academic Editor

PLOS ONE

---

## [Editor Report · Acceptance letter]

10 Jul 2023

PONE-D-23-05092R1 

Cost-Effectiveness Analysis of Adding Tuberculosis Household Contact Investigation on Passive Case-Finding Strategy in Southwestern Uganda 

Dear Dr. Odongo:

I'm pleased to inform you that your manuscript has been deemed suitable for publication in PLOS ONE. Congratulations! Your manuscript is now with our production department. 

Kind regards, 

on behalf of

Mr Hamufare Dumisani Dumisani Mugauri 

Academic Editor

PLOS ONE